# A New Approach for the Treatment of Recurrent Vulvovaginal Candidiasis with a Combination of Pea Protein, Grape Seed Extract, and Lactic Acid Assessed In Vivo

**DOI:** 10.3390/jof8121251

**Published:** 2022-11-27

**Authors:** Irene Paterniti, Giovanna Casili, Alessia Filippone, Marika Lanza, Alessio Ardizzone, Anna Paola Capra, Michela Campolo, Emanuela Esposito

**Affiliations:** Department of Chemical, Biological, Pharmaceutical and Environmental Sciences, University of Messina, Viale Ferdinando Stagno d’Alcontres 31, 98166 Messina, Italy

**Keywords:** recurrent vulvovaginal candidiasis, pea protein (PP), grape seed extract (GS), lactic acid (LA)

## Abstract

Background: Vulvovaginal candidiasis (VVC) is considered the second most common vaginal infection. Up to 8% of women in various populations experience more than three or four episodes within one year, which is regarded as recurrent vulvovaginal candidiasis (RVVC). Current therapies involve antifungal drugs that provide static effects but do not prevent recurrences due to increased antimicrobial resistance; thus, alternative therapies to antifungals are needed to prevent RVVC. Methods: A murine model of *Candida albicans*-induced RVVC was performed to evaluate the efficacy of a topical product containing pea protein (PP), grape seed extract (GS), and lactic acid (LA) to treat recurrent infections. Mice were inoculated with three separate vulvovaginal infections of 5 × 10^4^ cells/mL *C. albicans,* and histological evaluation, a myeloperoxidase (MPO) assay. and an ELISA kit for Prostaglandin E2 (PGE2) on vaginal tissues were performed. Results: The data obtained highlighted that the combination of PP, GS, and LA significantly preserved vaginal tissue architecture and prevented vaginal inflammation, proving its efficacy for the management of RVVC. Moreover, the combination of PP, GS, and LA notably increased azole efficacy by adding a new mechanism of action when administered concomitantly. Conclusion: Taken together, results demonstrated that the treatment with a combination of PP, GS, and LA is able to reduce the adhesion of *C. albicans.*

## 1. Introduction

Vulvovaginal candidiasis (VVC) is an exceedingly common mucosal infection of the lower female reproductive tract, caused mostly by a polymorphic opportunistic fungus. *Candida albicans*, along with other closely related *Candida* species, are the primary causative agents of VVC that represent the most prevalent human candida infection, estimated to afflict approximately 75% of all women at least once in their lifetime. Recurrent VVC (RVVC, defined as > 3 episodes per year) affects nearly 8% of women globally, especially in immunocompromised patients [1,2]. This is due to an over colonization of various *Candida* species in the vaginal lumen, which causes symptomatic inflammation of the vagina [3]. In its yeast form, *C. albicans* is well tolerated by the vaginal epithelium, but upon morphotype switching to the invasive hyphal form, co-regulated by genes encoding virulence factors such as secreted aspartyl proteases (Sap) and candidalysin, the tolerance threshold is surpassed, triggering intense inflammatory responses and tissue damage [4,5]. Generally, the most common predisposing factors for candidiasis are poor hygiene, nutritional deficiencies (iron, folic acid, and vitamin deficiency), and an age-related shift toward a carbohydrate-rich diet; furthermore, wearing mobile prosthetic replacements, such as catheters and intrauterine devices (IUDs), serves as a reservoir of yeasts and contributes to recurrent infection by *C. albicans*. In addition to pregnancy, some drugs such as antibiotics and systemic corticosteroids and concomitant diseases such as endocrinopathy and autoimmune diseases can cause infection [4,5]. Common disease symptoms include vaginal itching, vulvar edema, fissures, and excoriations, often accompanied by thick curdy, vaginal discharge [6] but also non-physiological symptoms such as depression, helplessness, and decreased quality of life [7]. According to the Clinical Practice Guidelines, VVC can be treated with topical or oral antifungals, of which azoles (miconazole, clotrimazole, and fluconazole) are the most commonly prescribed, but the static activity of azoles and inadequate immune-mediated clearance could represent key drivers of disease recurrence [8]. There is a marked increase in resistance of *C. albicans* to antifungal agents, causing multidrug resistance to emerge [9,10]. The rates of azole resistance are highly variable, and they may be influenced by the prescription patterns of clinicians for both prophylaxis and treatment purposes [11]. In particular, the maintenance therapy with azoles reduces the clinical recurrence rate during therapy in patients with RVVC, but there is usually no long-term remission. Moreover, there are well-characterized safety risks for fluconazole, including liver toxicity, drug interactions, and pregnancy warnings [12]. Consequently, alternative and/or complementary therapies are needed to effectively reduce the VVC and its associated recurrences. Much attention has been given to naturally derived ingredients in recent years. For instance, phytocompounds can help promote antifungal activity against candidiasis [13,14,15]. Pea protein (PP) is derived from the Pisum sativum plant, known for its film-forming features on diverse epithelia [16], which can help promote antifungal activity [17,18]. Interestingly, phenolic compounds, the most widely distributed class of natural molecules in plants, have displayed several biological properties including antifungal activity. It has been reported that flavonoids contained in grape seed extract (GS) may help promote an inhibitory effect on gram-positive and gram-negative bacteria and some Candida species through various mechanisms [19,20]. Moreover, GS polyphenols have a higher antioxidant activity compared to other well-known antioxidants, such as vitamin C, vitamin E, and β-carotene [21]. One of the most notable features of C*. albicans*-associated dysbiosis is a marked reduction of lactic acid (LA) levels. Lactobacilli species produce antimicrobial peptides and acidify the vaginal environment (pH  <  4.5) through lactic acid (LA) production to create a hostile environment for most pathogens, including *C. albicans* [22], and preserve the vaginal microbiota [23]. Therefore, the aim of this study was to evaluate the efficacy of a product containing PP, GS, and LA to treat recurrent infections in a murine model of *C. albicans*-induced RVVC.

## 2. Materials and Methods

### 2.1. Animals

The study was conducted in female CD-1 mice, aged 8 to 10 weeks (25–30 gr). The animals were housed in steel cages in a room maintained at 22 ± 1 °C with a 12-h dark and 12-h light cycle and provided with standard rodent food and water. The animals’ care has been approved by the Board of Auditors of the University of Messina and complies with the regulations in Italy (DM 116192) and Europe (European Directive 2010/63/EU amended by Regulation 2019/1010). The mice used for this study were selected from those suitable and available at that time.

### 2.2. Microorganisms and Growth Conditions

A clinically isolated strain of *C. albicans* (SC5314), purchased by ATCC, was used for vulvovaginal inoculations. SC5314 were grown in phyton peptone broth for 18 h at 25 °C on an orbital shaker at 7000 rpm. Stationary phase blastoconidia were adjusted to 5 × 10^4^ cells/mL. Each inoculum solution was prepared from freshly subcultured SC5314 on the day of inoculation. The viable count to reach the required number was made through a spread plate technique, as previously described [24].

### 2.3. Candida albicans-Induced Vaginitis Model

For the RVVC model, mice received three separate vulvovaginal infections of 5 × 10^4^ cells/mL *C. albicans* strain SC5314 via instillation into the vaginal lumen [25]. Infections were followed by four days of non-treatment, after which the treatment containing PP, GS, and LA was initiated for a duration of seven days. For the second and third rounds of infection, mice were re-inoculated with 5 × 10^4^ SC5314 cells/mL, 4 weeks after clearing the infection. Mice in the control group received inoculations of saline solutions and vaginal douches concurrently and in the same manner as the infected mice (Figure 1). The general conditions of the animals were monitored daily. The mice were sacrificed 74 days post-infection.

### 2.4. Experimental Groups

Mice were divided into the following experimental groups:Control animal: mice received inoculations of saline (no infections);Control + clotrimazole: mice received inoculations of clotrimazole (no infections);Control + therapeutic: mice received inoculations from the therapeutic group (no infections);RVVC: mice received three separate vulvovaginal infections with 5 × 10^4^ *C. albicans*;RVVC + clotrimazole: mice received three separate vulvovaginal infections with 5 × 10^4^ *C. albicans*; after each *C. albicans* inoculation, mice were treated with clotrimazole alone (for 3 days) + saline (7 days);RVVC + clotrimazole + therapeutic: mice received three separate vulvovaginal infections with 5 × 10^4^ *C. albicans*; after each *C. albicans* inoculation, mice were treated with clotrimazole alone (for 3 days) + therapeutic alone (for 7 days);RVVC + therapeutic: mice received three separate vulvovaginal infections with 5 × 10^4^ *C. albicans*; after each *C. albicans* inoculation, mice were treated with the therapeutic alone (for 7 days) in each round.

The minimum number of mice for every technique was estimated with the statistical test “ANOVA: Fixed effect, omnibus one-way” with G-power software. This statistical test generated a sample size equal to *n  *=  8 mice for each technique and *n* = 16 for each group.

### 2.5. Histological Evaluation

The collected vaginal tissues were fixed with 10% neutral formalin, dehydrated with graduated ethanol, and embedded in paraffin. Subsequently, the 7 μm thick tissue sections were deparaffinized with xylene and stained with hematoxylin and eosin to detect mucosal thickness and infiltration of inflammatory cells. An inverted microscope with two charge-coupled device (CCD) cameras (magnification: ×200; Nikon, Tokyo, Japan) was used to observe the colored sections.

### 2.6. Myeloperoxidase Assay

Myeloperoxidase activity (MPO), an index of polymorphonuclear cell accumulation, was determined in the vaginal tissues, as previously described [26]. Vaginal tissues collected at the specified time were homogenized in a solution containing 0.5% hexa-decyl-trimethyl-ammonium bromide dissolved in 10 mm potassium phosphate buffer (pH 7) and centrifuged for 30 min at 20,000× *g* at 4 °C. An aliquot of the supernatant was then allowed to react with a solution of tetra-methyl-benzidine (1.6 mm) and 0.1 mm H_2_O_2_. The rate of change in absorbance was measured spectrophotometrically at 650 nm using a Thermo Scientific™ Multiskan FC Microplate Spectrophotometer (Model: 51119100). MPO activity was expressed in units/mg protein.

### 2.7. ELISA Assay

To evaluate the inflammatory response after three separate vulvovaginal infections by *C. albicans* (RVVC), the levels of Prostaglandin E2 (PGE2) were measured in tissues collected after infection by enzyme-linked immunosorbent assay (ELISA), according to the manufacturer’s instructions. Briefly, samples were thawed on ice and homogenized in a specific lysis buffer; subsequently, the samples were homogenized and centrifuged. Supernatants were collected and stored at −20 °C. PGE2 quantity was measured using a microplate reader at 450 nm.

### 2.8. Materials

All chemicals were obtained from the highest grade of commercial sources. The product containing a combination of pea protein, grape seed extract and lactic acid, was kindly provided by DEVINTEC SAGL (Lugano, Switzerland).

### 2.9. Statistical Analysis

All values reported in the figures and in the text are indicated as mean ± standard deviation (SD) and are representative of at least three independent experiments. Results were analyzed by one-way ANOVA followed by a Bonferroni post-hoc test for multiple comparisons. A *p*-value of less than 0.05 was considered significant.

## 3. Results

### 3.1. Histological Evaluation of the Therapeutic’s Efficacy in RVVC

Candidiasis is characterized by specific histological patterns; particularly, vulvovaginal candidiasis provokes histologic lesions characterized by inflammatory infiltrate [27]. In this study, the histological examination of the vaginal tissue revealed characteristic pathological changes after three separate vulvovaginal infections of *C. albicans* (RVVC) (Figure 2C; see histological score Figure 2E) compared to the control groups (Figure 2A,B; see histological score Figure 2E). The product containing PP, GS, and LA significantly preserved vaginal tissue architecture following infections (RVVC + PP + GS + LA) (Figure 2D; see histological score Figure 2E). The low cellular toxicity of the therapeutic supported the in vivo observation that the product alone did not affect the morphology of vaginal tissues.

### 3.2. Protective Effects of the Therapeutic to Prevent Vaginal Inflammation Related to RVVC

MPO is a heme-containing peroxidase mainly expressed in neutrophils, that plays a pivotal role in inflammation by discharging a variety of lysosomal components and generating reactive oxygen metabolites; interestingly, neutrophils transform prostaglandins (PGs) by a MPO-dependent mechanism [28,29]. Significant oxidative stress and damage occurs extracellularly as a result of MPO release via phagolysosomal leakage and cell lysis. MPO binds with a higher affinity to some damaged extracellular matrix (ECM) components that act as chemoattractants, modulating MPO activity and altering cell function at sites of leukocyte infiltration and activation, with subsequent tissue damage and dysfunction. PGE2 is one of the most abundant PGs produced in the body and during inflammation. PGE2 is of particular interest because it is involved in all processes leading to the classic signs of phlogosis [30]. MPO activity and PGE2 quantity were measured in vaginal tissue, and both MPO and PGE2 levels were found to be significantly higher in the RVVC group compared to control groups (Figure 3A,B). Here, the treatment with the therapeutic containing PP, GS, and LA significantly reduced MPO activity (Figure 3A) and PGE2 quantity, respectively (Figure 3B).

### 3.3. Role of the Therapeutic in Enhancing the Efficacy of Clotrimazole in RVVC-Related Histological Damage

*C. albicans* infections formed a significant number of biotic biofilms on the vaginal epithelium, inducing histopathological changes. The elevated fungicidal activity of clotrimazole against hyphae plus clotrimazole-induced hyphae-to-yeast reversion may help to dampen acute vaginal infections by reducing the relative proportion of hyphae and thus shifting to a non-invasive commensal-like population [31]. The study was performed to better evaluate the action of PP + GS + LA in avoiding recurrent infection following clotrimazole treatment and alone. The histological examination of vaginal epithelium treated with clotrimazole showed significantly reduced pathological changes provoked by RVVC (Figure 4E,F; see histological score Figure 4E). The association of clotrimazole plus PP + GS + LA notably increased the antifungal activity of clotrimazole, better preserving vaginal epithelium following infections (Figure 4F; see histological score Figure 4E); however, PP + GS + LA treatment alone positively restored pathological changes (Figure 4G; see histological score Figure 4E).

### 3.4. Role of the Therapeutic in Enhancing the Efficacy of Clotrimazole in RVVC-Related Vaginal Inflammation

RVVC is frequently accompanied by an inflammatory response in association with neutrophil infiltration, contributing to the symptoms [32]. Neutrophils are normally the first responders to acute inflammation with the potential to directly inflict tissue damage, driving inflammation through antigen presentation and the secretion of prostaglandins and cytokines [33]. In this study, MPO activity and PGE2 quantity were investigated in vaginal tissue, observing a notable decrease of both MPO and PGE2 in the RVVC + clotrimazole group compared to RVVC group (Figure 5A,B). Indeed, the treatment with clotrimazole in association with the therapeutics containing PP, GS, and LA reduced MPO activity (Figure 5A) and PGE2 quantity (Figure 5B) significantly, compared to treatment with clotrimazole alone.

## 4. Discussion

VVC represents the most prevalent fungal infections in humans; however, it is underestimated and regarded as an easy-to-treat condition [34]. RVVC is a difficult-to-manage condition that affects 5–8% of women of reproductive age [2]. The most common drugs for the clinical treatment of VVC include nystatin, fluconazole, and miconazole, but the complexity of RVVC physiopathology often leads to a gradual increase in drug resistance during the course of treatment [35]. Also, these treatments can lead to undesirable side effects, like gastrointestinal adverse reactions (nausea, vomiting, headaches, rash, abdominal pain, and diarrhoea) [36] and toxicity [37]. Therefore, the search for new safe and effective therapeutics with low toxicity is warranted [35]. In the last years, the antimicrobial activity of natural products has encouraged the development of alternative treatments. Recently, our attention has been focused on PP and GS for their peculiar properties in a murine model of VVC, demonstrating their ability to enhance the antifungal activity of fluconazole [38]. Furthermore, LA represents a major antimicrobial factor produced by Lactobacilli which supports the presence of the lactic acid bacteria. Lactobacilli are members of the microbiota of several human niches, including the vagina, with the capacity to suppress filamentation, a key virulence feature of *C. albicans* [39,40]. Therefore, the aim of this study was to evaluate the efficacy of a product containing PP, GS, and LA to treat recurrent infections in a murine model of *C. albicans*-induced RVVC. This experimental model was established according to previous reports [25], and the aforementioned expansion of this model favors a more comprehensive understanding of the mechanisms controlling VVC, providing important insights into the relationship between the widely used rodent models and human disease [41]. The aim was also to demonstrate the capacity of PP, GS, and LA to enhance the antifungal activity provided by the clotrimazole in RVVC.

RVVC predominantly causes spongiotic changes in the epidermis with irregular acanthosis, mild spongiosis, and intense itchiness of the vagina [27,42]. In this study, intravaginal treatment with the therapeutic containing PP, GS, and LA significantly preserved vaginal tissue architecture following three separate vulvovaginal infections of *C. albicans.* Furthermore, the association of clotrimazole plus therapeutics, containing PP, GS, and LA notably increased the antifungal activity of clotrimazole, significantly preserving vaginal epithelium integrity thanks to the muco-protective film that counteracts the recurrent infections.

Symptomatic RVVC is strongly associated with an acute inflammatory response characterized by polymorphonuclear neutrophils (PMNs) recruitment into the vaginal lumen [43]. In this study, treatment with the therapeutic significantly reduced vaginal PMN migration, indicating decreased MPO activity.

*C. albicans* is a ubiquitous fungal symbiont that resides on diverse human barrier surfaces, and both mammalian and fungal cells can convert arachidonic acid into the lipid mediator PGE2, but the physiological significance of fungus-derived PGE2 remains elusive [44]. Fungal production of PGE2 was associated with enhanced fungal survival within phagocytes, suggesting that *C. albicans* has evolved the capacity to produce PGE2 from arachidonic acid, a host-derived precursor, to promote its own colonization [45]. In this study, we demonstrated the capacity of the therapeutics containing PP, GS, and LA to reduce PGE2 quantity. Moreover, the co-administration of the therapeutic and clotrimazole significantly lowered PGE2 production, suggesting an enhancement of the clotrimazole efficacy. This may suggest the possibility of reducing the quantity of azoles used to treat the infection, consequently reducing the risk of developing antimicrobial resistance and azole-associated side effects.

Considering these compounds have the capacity to create a protective mechanical barrier on the vaginal mucosa, it would be intriguing to investigate their therapeutic effect on other fungal infections. In addition, the synergistic combination of these compounds may reduce the risk of Trichomonas vaginalis and gonorrhea infections, thus opening new horizons for an application in the pathophysiology of sexually transmitted diseases (STDs). Due to the complexity of the pathology, this study faced some limitations. Firstly, in vivo candidiasis susceptibility does not uniformly predict clinical success in patients; therefore, the translational significance of the efficacy of fungal elimination via biofilm inhibition could be investigated in the clinics. Moreover, the different vaginal microbiota composition between mice and humans [46] represents a further limitation of the study because the beneficial effects of natural compounds may undergo slight variations due to different successful Candida colonizations on different human and mouse endogenous flora.

Thirdly, limitations and clinical pitfalls in the clinical therapy of these natural compounds derive from their bioavailability and low absorption. Therefore, future research on nanoformulations for ointments or gels as versatile drug delivery systems would be needed. Nonetheless, PP, GS, and LA were able to suppress several major virulence factors of *C. albicans*, such as the ability to switch from yeast to mycelial form and the capacity to express several aspartyl proteases, suggesting that the synergy between mechanical effects and antifungal effects was more effective. The capacity of this therapeutic to inhibit the adherence of Candida has been directly correlated with the cell surface hydrophobicity of natural compounds (PP, GS, and LA) since these characteristics directly influence its adhesion to the epithelial tissue, forming a mechanical barrier against Candida. Indeed, further studies are necessary to elucidate other specific mechanisms of this therapeutic, which contains PP, GS, and LA as antifungal agents.

## 5. Conclusions

New therapeutic options and strategies are needed to address the challenges of azole resistance, as unnecessary use of antifungals can lead to antimicrobial resistance by selecting less-sensitive species. Our results suggest that the therapeutic containing PP, GS, and LA significantly preserves vaginal tissue architecture, preventing vaginal inflammation. Moreover, overall results demonstrate that the product significantly increases azoles’ efficacy by adding a new mechanism of action when administered concomitantly. In conclusion, we can affirm that PP, GS, and LA could represent valid alternative therapeutics for the management of RVVC.

## Figures and Tables

**Figure 1 jof-08-01251-f001:**
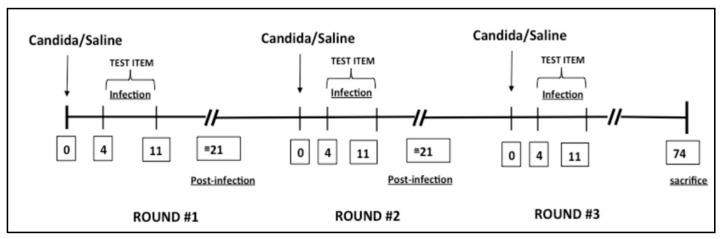
Experimental design.

**Figure 2 jof-08-01251-f002:**
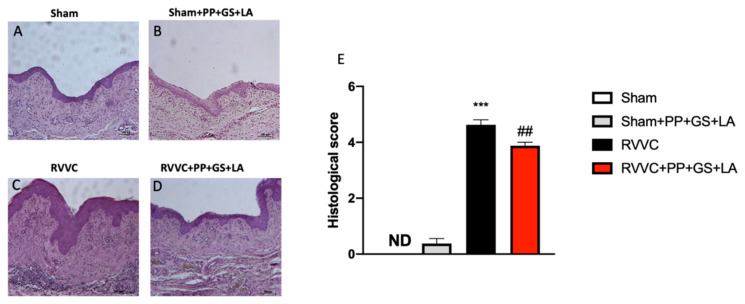
Histological evaluation of vaginal tissues. H/E staining was performed to evaluate the histological analysis of vaginal tissue. (**A**) Control mice; (**B**) Control mice treated with the therapeutic; (**C**) mice subjected to re-infections (RVVC); (**D**) mice subjected to RVVC and treated with the therapeutic; (**E**) Histological score. Values are indicated as the mean ± SD. *** *p* < 0.001 vs. Sham; ## *p* < 0.01 vs. RVVC. (**E**) *p*-value = 0,20.

**Figure 3 jof-08-01251-f003:**
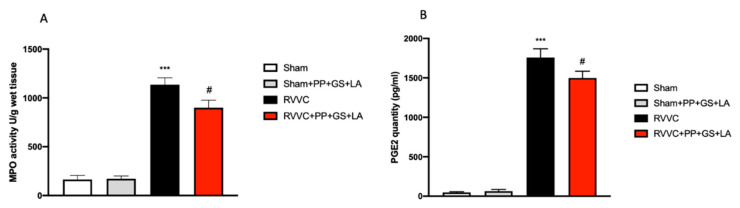
Myeloperoxidase (MPO) activity and Prostaglandin E2 (PEG2) quantity evaluation of vaginal tissues. MPO activity (**A**) and PGE2 quantity (**B**) were evaluated in vaginal tissues. Values are presented as the mean ± SD. *** *p* < 0.001 vs. Sham; # *p* < 0.05 vs. RVVC. (**A**) *p*-value = 0.74; (**B**) *p*-value = 0.29.

**Figure 4 jof-08-01251-f004:**
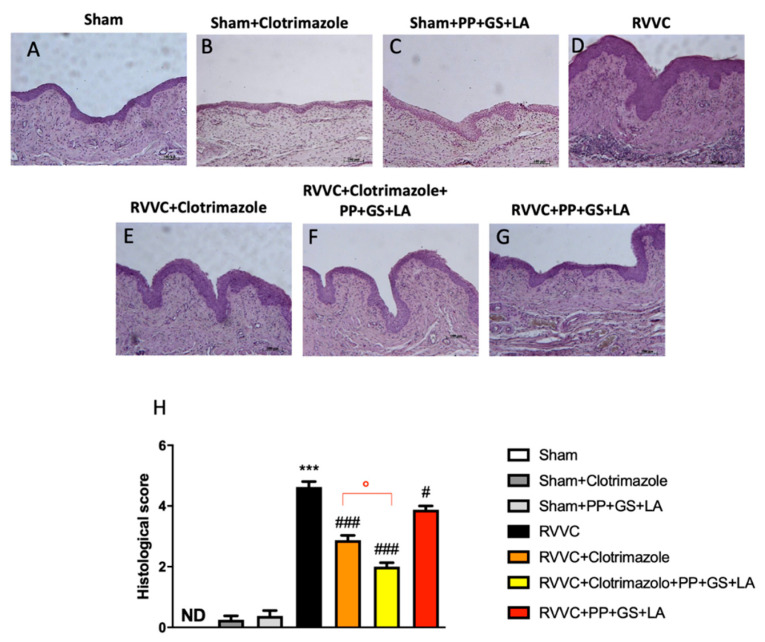
Histological evaluation of vaginal tissues. H/E staining was performed to evaluate the histological analysis of vaginal tissue. (**A**) Control mice; (**B**) Control mice treated with clotrimazole; (**C**) Control mice treated with the therapeutic; (**D**) mice subjected to re-infections (RVVC); (**E**) Mice subjected to RVVC and treated with clotrimazole; (**F**) Mice subjected to RVVC and treated with clotrimazole and PP + GS + LA; (**G**) Mice subjected to RVVC and treated with PP + GS + LA; (**H**) Histological score. Values are presented as the mean ± SD. *** *p* < 0.001 vs. Sham; ### *p* < 0.001 vs. RVVC and # *p* < 0.05 vs. RVVC; ⁰ *p* < 0.05 vs. RVVC + clotrimazole. (**H**) *p*-value = 0.42.

**Figure 5 jof-08-01251-f005:**
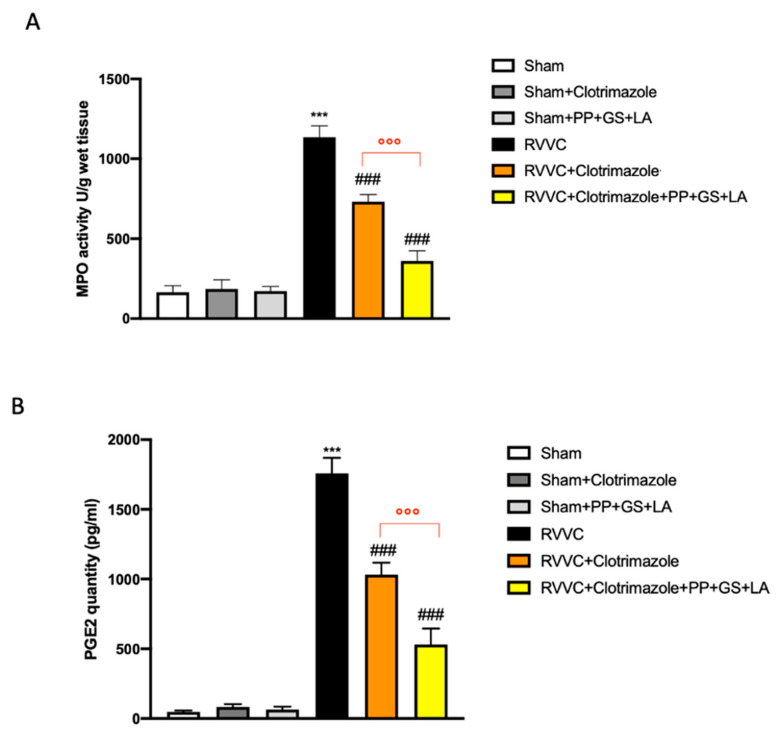
MPO activity and PEG2 quantity evaluation of vaginal tissues. MPO activity (**A**) and PGE2 quantity (**B**) were evaluated in vaginal tissues. Values are presented as the mean ± SD. *** *p* < 0.001 vs. Sham; ### *p* < 0.001 vs. RVVC. ⁰⁰⁰ *p* < 0.001 vs. RVVC + clotrimazole. (**A**) *p*-value= 0.94; (**B**) *p*-value = 0.49.

## Data Availability

Not applicable.

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
