# Peer review of "A New Approach for the Treatment of Recurrent Vulvovaginal Candidiasis with a Combination of Pea Protein, Grape Seed Extract, and Lactic Acid Assessed In Vivo"

_jof, 2022, doi:10.3390/jof8121251_

Round 1

Reviewer 1 Report

The manuscript needs extensive editing of the English language and style of presentation. The abstract needs moderate revision as the aim of the study was the only content of the methods, while the methodology was not appropriately summarized.  

Candida albicans was not italicized throughout the manuscript. The content of the introduction was adequate but needs grammatical revision.  There is a need to know the in vitro antifungal efficacy of the compounds.

The abstract was poorly structured and presented. The aim of the study was the only content of the methods, while the methodology was not appropriately summarized. The results contained conclusive remarks.

Introduction:  'Wearing a mobile prosthetic replacement' was stated as a risk factor, It would have been appropriate to cite examples. 

Methods: The units of the colony should be 5 X 104 cells/ml 'not' 5 X 104 cells/ml.

Results: There was no evidence of statistical power and analysis in the results

 Discussion: The content was adequate, but the readers could be misled because of the poor editing.

Author Response

The manuscript needs extensive editing of the English language and style of presentation. The abstract needs moderate revision as the aim of the study was the only content of the methods, while the methodology was not appropriately summarized.  

As suggested by reviewer, the authors edited the manuscript for English language and style of presentation. Furthermore, the authors improved the Abstract to better highlight the aim of the study, appropriately summarizing the methodology.

Candida albicans was not italicized throughout the manuscript. The content of the introduction was adequate but needs grammatical revision.  There is a need to know the in vitro antifungal efficacy of the compounds.

As suggested by reviewer, the authors italicized the term Candida Albicans throughout the manuscript. Also, the authors revised the Introduction section for grammatical revisions. Regarding the in vitro antifungal efficacy of the compounds, the authors previously demonstrated the antifungal activity of pea protein and grape seed on C. albicans proliferation, as showed in “Effect of pea protein plus grape seed dry extract on a murine model of Candida albicans induced vaginitis” by Esposito E. et al., 2018. Regarding the lactic acid, the other compound used in this study, it is known that lactic acid create a hostile environment for most pathogens, including C. albicans, as showed in “Interplay between Candida albicans and Lactic Acid Bacteria in the Gastrointestinal Tract: Impact on Colonization Resistance, Microbial Carriage, Opportunistic Infection, and Host Immunity” by Zeise KD et al., 2021 and in “Effect of Acetic Acid and Lactic Acid at Low pH in Growth and Azole Resistance of Candida albicans and Candida glabrata” by Lourenco A. et al., 2019.

The abstract was poorly structured and presented. The aim of the study was the only content of the methods, while the methodology was not appropriately summarized. The results contained conclusive remarks.

As suggested by reviewer, the authors improved the Abstract to better highlight the aim of the study, appropriately summarizing the methodology.

Introduction:  'Wearing a mobile prosthetic replacement' was stated as a risk factor, It would have been appropriate to cite examples. 

As suggested by reviewer, the authors cited appropriated examples regarding the risk factors related to “wearing a mobile prosthetic replacement”; particularly, the authors highlighted that pathogenic microbes can form biofilms on the inert surfaces of implanted devices such as catheters and intrauterine devices (IUDs). Indeed, the presence of biofilm on the patients' IUDs serves as a reservoir of yeasts and contributed to recurrent infection by Candida albicans. This fact can be verified by the observation of yeasts in the biofilm through electron scanning microscopic analysis, suggesting the possibility that the majority of these microorganisms had been present on these surfaces for possibly a long time (“Biofilm formation on intrauterine devices in patients with recurrent vulvovaginal candidiasis” by Auler ME. Et al.,  2010).

Methods: The units of the colony should be 5 X 104 cells/ml 'not' 5 X 104 cells/ml.

As suggested by reviewer, the authors corrected the unit of colony, superscripting 104.

Results: There was no evidence of statistical power and analysis in the results.

As suggested by reviewer, the authors added in Materials and methods (section 2.4) that a statistical power was performed. Specifically, the minimum number of mice for every technique was estimated with the statistical test “ANOVA: Fixed effect, omnibus one-way” with G-power software. This statistical test generated a sample size equal to n = 8 mice for each technique and n = 16 for each group. Furthermore, as showed in section 2.9, the authors indicated that all values reported in the figures and in the text are expressed as mean ± standard deviation (SD)  and are representative of at least three independent experiments. Results were analyzed by one-way ANOVA followed by a Bonferroni post-hoc test for multiple comparisons. A p-value of less than 0.05 was considered significant. Furthermore, to increase the statistical power, the authors specified p value for each result showed in figure legends.

Discussion: The content was adequate, but the readers could be misled because of the poor editing.

As suggested by reviewer, the authors better edited the Discussion section, to better highlight the content of the research.

Reviewer 2 Report

Using an animal model, the authors analyzed the effect of natural compounds (as a topical product) on a cure for recurrent vulvovaginal candidiasis. They investigated the efficacy of clotrimazole in combination with natural compounds.  It is a well-written paper with a clearly presented experimental part with an animal model described by the authors in their earlier article. Some comments are shown below:

Line 44: C.albicans is not the only Candida species that cause VVC. Please, revise the sentence.

Line 96: who/where/ the C.albicans strain has been isolated? From what source was the strain extracted? If available, please, provide an appropriate reference.

Discussion: The authors discussed the limitations of the study, however, they haven’t mentioned that the mouse vaginal microbiota is diverse from the human vaginal microbiota that may impact the activity of natural compounds.

Lines 98, 103, 107, 116, and elsewhere in the text: 5x104, should be a superscript of 4.

Lines 106-107. The sentence is unclear: what do the authors mean “4 weeks after clearing the infection”?

Author Response

Using an animal model, the authors analyzed the effect of natural compounds (as a topical product) on a cure for recurrent vulvovaginal candidiasis. They investigated the efficacy of clotrimazole in combination with natural compounds.  It is a well-written paper with a clearly presented experimental part with an animal model described by the authors in their earlier article. Some comments are shown below:

Line 44: C. albicans is not the only Candida species that cause VVC. Please, revise the sentence.

As suggested by reviewer, the authors revised the sentence, specifying that VVC was caused by an over colonization of various Candida species in  involved in VVC development.

Line 96: who/where/ the C. albicans strain has been isolated? From what source was the strain extracted? If available, please, provide an appropriate reference.

  1. albicans strain used (SC5314 / ATCC MYA-2876) was purchased by ATCC.

Discussion: The authors discussed the limitations of the study, however, they haven’t mentioned that the mouse vaginal microbiota is diverse from the human vaginal microbiota that may impact the activity of natural compounds.

As suggested by reviewer, the authors improved the Discussion section, better highlighting the limitations related to the different vaginal microbiota between mouse and human. Indeed, in humans, the vaginal microbiota exists in five distinct community state types (CSTs) which are generally dominated by Lactobacillus spp, while in mouse, sequencing analysis revealed five distinctive community states of the vaginal microbiota dominated largely by Staphylococcus and/or Enterococcus, Lactobacillus or a mixed population, that are relatively unstable in mice (“The murine vaginal microbiota and its perturbation by the human pathogen group B Streptococcus” by Vrbanac A et al., 2018). Therefore, a further limitation of the study could be that the beneficial effects of natural compounds may undergo slight variations due to a different successful Candida colonization on different human and mice endogenous flora.

Lines 98, 103, 107, 116, and elsewhere in the text: 5x104, should be a superscript of 4.

As suggested by reviewer, the authors corrected the unit of colony, superscripting 104.

Lines 106-107. The sentence is unclear: what do the authors mean “4 weeks after clearing the infection”?

The authors mean that mice are capable of clearing Candida infections, usually within 4 weeks, thus recurring vaginosis is reproduced every 4 weeks.

Round 2

Reviewer 1 Report

Extensive revision of the manuscript was noted, and it's now clearer and more enjoyable to read. 

However, kindly rephrase lines 265 and 266.

Thank you.